# Ratiometric Near-Infrared Fluorescence Liposome Nanoprobe for H_2_S Detection In Vivo

**DOI:** 10.3390/molecules28041898

**Published:** 2023-02-16

**Authors:** Luyan Wu, Yili Liu, Junya Zhang, Yinxing Miao, Ruibing An

**Affiliations:** 1Jiangsu Key Laboratory for Biosensors, Key Laboratory for Organic Electronics and Information Displays, Institute of Advanced Materials (IAM), Jiangsu National Synergetic Innovation Center for Advanced Materials (SICAM), Nanjing University of Posts and Telecommunications, Nanjing 210023, China; 2State Key Laboratory of Analytical Chemistry for Life Science, Chemistry and Biomedicine Innovation Center (ChemBIC), School of Chemistry and Chemical Engineering, Nanjing University, Nanjing 210033, China; 3Institute of Optical Functional Materials for Biomedical Imaging, School of Chemistry and Pharmaceutical Engineering, Shandong First Medical University & Shandong Academy of Medical Science, Taian 271016, China

**Keywords:** near-infrared fluorescence, hydrogen sulfide (H_2_S), activatable imaging, ratiometric imaging, fluorescence imaging

## Abstract

Accurate detection of H_2_S is crucial to understanding the occurrence and development of H_2_S-related diseases. However, the accurate and sensitive detection of H_2_S in vivo still faces great challenges due to the characteristics of H_2_S diffusion and short half-life. Herein, we report a H_2_S-activatable ratiometric near-infrared (NIR) fluorescence liposome nanoprobe HS-CG by the thin-film hydration method. HS-CG shows “always on” fluorescence signal at 816 nm and low fluorescence signal at 728 nm; the NIR fluorescence ratio between 728 and 816 nm (F_728_/F_816_) is low. Upon reaction with H_2_S, the fluorescence at 728 nm could be more rapidly turned on due to strong electrostatic interaction between enriched HS^−^ and positively charged 1,2-dihexadecanoyl-sn-glycero-3-phosphocholine (DPPC) doped in the liposome nanoprobe HS-CG, resulting in a large enhancement of F_728_/F_816_, which allows for sensitive visualization of the tumor H_2_S levels in vivo. This study demonstrates that this strategy of electrostatic adsorption between HS^−^ and positively charged molecules provides a new way to enhance the reaction rate of the probe and H_2_S, thus serving as an effective platform for improving the sensitivity of imaging.

## 1. Introduction

Hydrogen sulfide (H_2_S), the third endogenous gaseous transmitter besides nitric oxide (NO) and carbon monoxide (CO), is produced from a cysteine substrate or its derivatives and catalyzed by several enzymes such as cystathionine g-lyase (CSE), cystathionine b-synthase (CBS), and 3-mercaptopyruvate sulfurtransferase [1,2,3]. H_2_S plays an important role in biological functions and mediating many biological processes including cytoprotection [4,5], mediation of neurotransmission [6,7], vascular tone regulation [8,9], anti-inflammation effect [10,11], etc. However, abnormal levels of H_2_S are correlated with the symptoms of numerous diseases such as Down syndrome, Alzheimer’s disease, and cancers [12,13,14,15,16]. Therefore, the highly sensitive and specific accurate detection of H_2_S levels in biological systems is of great significance for understanding H_2_S relevant physiological and pathological processes.

Ratiometric fluorescence imaging is a method where intensities at two or more different wavelengths are measured to detect the dynamic behavior and function relationships of biomolecules in living organisms [17,18,19]. Ratiometric fluorescence imaging provides built-in self-calibration for factors unrelated to the analyte (e.g., variable probe concentration and distribution, instrument sensitivity, dynamic biological environment and photobleaching), thus improving the reliability of biomolecular detection [20,21,22,23]. Based on the ratiometric imaging strategy, many ratiometric fluorescence imaging probes have been developed to visualize the spatiotemporal distribution of fluoride anions [24], hypochlorous acid (HClO) [25,26], and H_2_S [27,28,29,30,31,32], which have greatly improved the accuracy of molecular detection and imaging. However, because most of the analytical wavelengths of ratiometric fluorescence imaging are in the visible light region, this suffers from significant absorption, scattering, and spontaneous fluorescence interference induced by biological tissues, resulting in extremely low tissue penetration, which limits their application in vivo. Ratiometric near-infrared (NIR) fluorescent imaging probes have attracted wide attention and have been successfully applied to the detection of various analytes [33], which reduces the absorption and scattering of light by tissues caused by the visible light region, and improves the tissue penetration depth of ratiometric fluorescence imaging. In particular, ratiometric NIR fluorescence imaging probes have been developed to detect H_2_S levels [30,34,35,36,37]. Nevertheless, most of them are still very limited for in vivo imaging because one of the analytical wavelengths is in the visible light region and the reaction rate with H_2_S is low. Considering the above situations, ratiometric fluorescence probes with a fast response toward H_2_S, enhanced sensitivity, and NIR analytical wavelength are still necessary for the precise measurement of H_2_S levels in living systems.

Herein, we report an activatable ratiometric NIR fluorescence liposome nanoprobe (HS-CG) for the detection of H_2_S in the tumors of mice. HS-CG displays an “always-on” NIR fluorescence signal at 816 nm and turns on the NIR fluorescence signal at 728 nm upon rapid reaction with H_2_S. Therefore, the fluorescence intensity ratio between 728 and 816 nm (F_728_/F_816_) increases significantly when the HS-CG rapidly reacts with H_2_S, permitting the sensitive and reliable monitoring of endogenous H_2_S dynamic changes in vivo.

## 2. Results and Discussion

### 2.1. Design of Ratiometric NIR Fluorescence Liposome Nanoprobe for H_2_S Detection

Figure 1 shows the design of the NIR ratiometric fluorescence liposome nanoprobe (HS-CG). HS-CG was prepared by 1,2-dihexadecanoyl-sn-glycero-3-phosphocholine (DPPC)-, cholesterol-, and 1,2-distearoyl-sn-glycero-3-phosphoethanolamine-N-[methoxy(polyethylene glycol)-2000] (DSPE-mPEG2000)-assisted encapsulating of H_2_S activatable IR-HS and indocyanine green (ICG) through thin-film hydration method. IR-HS is used as an imaging reagent because we previously proved that it has excellent stability and can produce bright NIR fluorescence at 728 nm under 680 nm excitation when IR-HS reacts with H_2_S rapidly (Figure 1a) [38]. ICG is used as a fluorescent dye because of its stable and bright NIR fluorescence at 816 nm [39,40], which can avoid interference from the fluorescence of dye 1 at 728 nm and use as a reliable internal reference for H_2_S detection. Initially, the liposome nanoprobe HS-CG displays the “always on” bright NIR fluorescence signal of ICG at 816 nm and the “off” fluorescence signal of IR-HS at 728 nm. The fluorescence intensity ratio between 728 and 816 nm (F_728_/F_816_) is very low. Upon reaction with H_2_S, IR-HS within HS-CG is converted into fluorescence dye 1 with fluorescence emission at 728 nm, and ICG’s fluorescence at 816 nm is unchanged, resulting in an increase in the fluorescence ratio F_728_/F_816_, which would be suitable for detecting H_2_S levels. The ratiometric NIR fluorescence liposome nanoprobe HS-CG possesses many unique advantages: (1) Due to the doping of DPPC, the positive charge on the surface of liposome nanoparticles HS-CG can effectively adsorb HS^−^ with a negative charge, which improves the reaction rate between IR-HS within HS-CG and H_2_S; (2) the “always on” NIR fluorescence signal of ICG at 816 nm of the liposome nanoparticles HS-CG prepared by DPPC and DSPE-mPEG2000 is significantly higher than that of micellar nanoparticles HS-CG NP prepared by DSPE-mPEG2000, which improves the sensitivity of tumor detection; and (3) the NIR fluorescence intensity ratio F_728_/F_816_ provided by HS-CG can avoid the interference of factors unrelated to the analyte such as the concentration of the HS-CG and environmental factors, and improve the reliability and sensitivity of H_2_S detection. With these advantages of liposome nanoparticles HS-CG, we used the “always on” fluorescence signal at 816 nm to noninvasively track the delivery and accumulation of HS-CG in the tumors after systemic administration while the activatable fluorescence signal at 728 nm was used to reliably detect tumor endogenous H_2_S in mice.

### 2.2. Synthesis and Characterization of Liposome Nanoprobe HS-CG

We first synthesized the liposome nanoprobe HS-CG by the thin-film hydration method. UV–Vis–NIR adsorption spectra demonstrated characteristic absorption peaks of ICG and IR-HS, with encapsulation efficiencies of nearly 100%. The mass ratio of DPPC, cholesterol, DSPE-mPEG2000, ICG, and IR-HS was 4:2:4:0.035:0.027. These results proved the successful preparation of the liposome nanoprobe HS-CG (Figure 2a). Dynamic light scattering (DLS) analysis demonstrated that liposome nanoprobe HS-CG could well disperse in an aqueous solution. The hydrodynamic size of the liposome nanoprobe HS-CG was found to be ~100 nm (Figure 2b). Transmission electron microscopy (TEM) analysis showed that the nanoprobe HS-CG appeared as a vesicle shape (Figure 2b, inset). UV–Vis–NIR adsorption spectra demonstrated that the absorption of HS-CG between 462 and 636 nm declined, and the absorption between 636 nm and 752 nm obviously increased after the reaction with NaHS (Appendix A). Moreover, the fluorescence spectra of HS-CG initially had a NIR fluorescence emission at 816 nm (Figure 2c). Upon reaction with NaHS, the fluorescence of the HS-CG at 728 nm was significantly enhanced by ~53-fold, while the fluorescence of the HS-CG at 816 nm remained unchanged. To verify the role of DPPC in the liposome nanoprobe HS-CG, we prepared the micellar nanoparticles (HS-CG NPs) with DSPE-PEG-assisted encapsulation of ICG and IR-HS. DLS showed that the particle size of the HS-CG NP was ~52 nm, and the TEM showed that the HS-CG NPs had a spherical micelle morphology (Figure 2d). Fluorescence spectra showed that the HS-CG NPs had a fluorescence peak at 816 nm. When HS-CG NP reacted with NaHS, the fluorescence at 728 nm increased by 36-fold (Figure 2e), which indicated that the fluorescence activation ratio at 728 nm of the HS-CG NPs was significantly lower than that of the liposome nanoparticles HS-CG. Moreover, it is noteworthy that the fluorescence intensity of the micellar nanoparticles HS-CG NPs at 816 nm was 1.8-fold lower than that of the liposome nanoparticles HS-CG. Additionally, we showed that when the liposome nanoparticles HS-CG and micelle nanoparticles HS-CG NP with the same concentration were incubated with NaHS (1 mM), the fluorescence intensity of the liposome nanoparticles HS-CG reached a plateau at 200 s, 141 s earlier than that of the micelle nanoparticles HS-CG NP (Figure 2f), which was due to the positively charged surface of the liposome nanoparticles HS-CG with doping DPPC possibly accelerating the reaction between HS-CG and H_2_S. In addition, we found that the H_2_S-activated increment of the fluorescence ratio (F_728_/F_816_) could be similarly acquired in different pH PBS buffers (6.0–8.0) (Appendix A). Taken together, liposome nanoparticles HS-CG could rapidly react with H_2_S, and the fluorescence signal at 728 nm was significantly enhanced, thus effectively improving the sensitivity of H_2_S detection.

### 2.3. Response of Liposome Nanoprobe HS-CG toward H_2_S In Vitro

To detect the sensitivity of liposome nanoprobe HS-CG toward H_2_S, the fluorescence spectra of HS-CG after incubation with different concentrations of NaHS (0, 1, 2, 5, 10, 15, 20, 50, 80, 100, 150, 200, 300, 400, 500 μM) for 6 min were acquired. Figure 3a–d showed that the fluorescence of HS-CG at 728 nm increased with the concentration of NaHS, while the fluorescence at 816 nm remained unchanged. The fluorescence ratio (F_728_/F_816_) versus NaHS concentration from 0 to 20 μM showed a good linear correlation; the detection limit (LOD) of HS-CG toward H_2_S was determined to be ~0.13 μM (Figure 3e). To evaluate the selectivity of HS-CG toward H_2_S, HS-CG was incubated with NaHS and other reducing agents (e.g., GSH, VC, _L_-Cys, Hcy). Figure 3f–i demonstrates that NaHS and other reductants (e.g., GSH, VC, _L_-Cys, Hcy) containing NaHS could activate the fluorescence of HS-CG at 728 nm, which resulted in a significant enhancement in the fluorescence ratio F_728_/F_816_ (Figure 3h). However, the fluorescence spectra and fluorescence ratio F_728_/F_816_ were unchanged for HS-CG following treatment with the other reductants, showing that the liposome nanoprobe HS-CG was highly specific for H_2_S. Additionally, we showed that liposome HS-CGs were quite stable in PBS buffer (pH 7.4) for 5 days, with little change in the fluorescence of IR-HS and ICG and fluorescence ratio F_728_/F_816_ upon incubation (PBS, pH 7.4) for 120 h (Appendix A). These results demonstrate that liposome HS-CG could serve as an efficient ratiometric imaging probe in vivo.

### 2.4. Imaging of H_2_S in Tumor Cells

Prior to cell ratiometric imaging, the cell viability of the liposome nanoprobe HS-CG was first evaluated via the MTT analysis. Appendix A shows that HS-CG had excellent biocompatibility for cells during the incubation of HCT116 colorectal cancer cells or human embryonic kidney cells (HEK293 cells) with HS-CG for 24 h. To optimize the incubation time and concentration, we then utilized the “always on” fluorescence of ICG within HS-CG to examine the uptake of HS-CG in the HCT116 cells. We showed that the fluorescence of ICG in cells increased in a time- and concentration-dependent manner (Appendix A). The ICG fluorescence reached the maximum in cells following incubation with liposome nanoprobe HS-CG at a concentration of 4 μM for 3 h. Moreover, we found that the fluorescence of HS-CG could well coincide with the fluorescence of lysosomes in cells, indicating that HS-CG could effectively locate the lysosome of the cells (Appendix A).

With the optimized incubation conditions, the ratiometric fluorescence imaging of H_2_S in living cells was conducted following incubation with HS-CG (4 μM, 3 h). Figure 4a–c shows that HCT116 cells incubated with HS-CG possessed bright fluorescence under both fluorescence dye 1 and the ICG channels, and the intracellular fluorescence ratio F_IR-HS_/F_ICG_ was measured was ~0.21 ± 0.07. When ZnCl_2_ (an H_2_S scavenger) was added to the cells, the intracellular fluorescence of IR-HS was reduced, while the fluorescence of ICG remained unchanged. Therefore, the F_IR-HS_/F_ICG_ was reduced to ~0.06 ± 0.03. When NaHS was added to cells to upregulate the concentration of exogenous H_2_S in the cells, the F_IR-HS_/F_ICG_ was obviously enhanced to ~1.0 ± 0.1 because the fluorescence of IR-HS was activated, which was decreased by ZnCl_2_. HS-CG was then applied to detect endogenous H_2_S generation in HCT116 cells. Upon the incubation with _L_-Cys (endogenous substrate of CSE and CBS for H_2_S production), the fluorescence of IR-HS in the cells became brighter and the F_IR-HS_/F_ICG_ was obviously ~4.6-fold higher than that of the untreated cells (Control group), which could be effectively blocked by the CSE inhibitor propargylglycine (PAG) and CBS inhibitor aminooxyacetic acid (AOAA) (Figure 4d). These image results show that HS-CG could monitor the H_2_S levels in cells via ratiometric fluorescence imaging.

### 2.5. Imaging of H_2_S in Tumor Cells

Encouraged by the results of cell imaging, we then utilized the HS-CG to conduct ratiometric fluorescence imaging for reporting the exogenous H_2_S levels in vivo. Mice with subcutaneous (s.c.) injection of HS-CG showed dark fluorescence at the IR-HS channel, and bright fluorescence at the ICG channel (Figure 5a,b). The fluorescence ratio F_IR-HS_/F_ICG_ was low (Figure 5c). However, the fluorescence signal at the IR-HS channel was significantly enhanced in living mice with s.c. injection of HS-CG plus NaHS, while ICG fluorescence remained unchanged, resulting in the F_IR-HS_/F_ICG_ ratio obviously being enhanced by ~5.1-fold (Figure 5a–c). When ZnCl_2_ was s.c. injected into mice treated with NaHS, the fluorescence of IR-HS decreased, with the F_IR-HS_/F_ICG_ reduced by ~1.9-fold (Figure 5a–c). These results indicate that HS-CG could serve as a ratiometric fluorescence imaging nanoprobe for monitoring exogenous H_2_S production in vivo.

When HS-CG was intravenous (i.v.) injected into s.c. HCT116 tumor-bearing mice, the “always-on” ICG fluorescence images showed that HS-CG could gradually accumulate in tumors (Appendix A). Fluorescence intensities gradually increased with time and reached the maximum at 12 h (Appendix A). Ex vivo fluorescence images of the main organs and tissues at 12 h showed that the fluorescence of the tumors was significantly higher than that of other organs (e.g., heart, liver, spleen, lung, kidneys, intestines, stomach), which revealed that the HS-CG mainly accumulated in tumor at 12 h (Appendix A). Moreover, the biodistribution analysis also indicated that HS-CG mainly accumulated in the tumor (%ID/g ≈ 17.7%) and liver (%ID/g ≈ 10.2%) at 12 h post i.v. injection into s.c. HCT116 tumor-bearing mice (Appendix A). Next, we utilized HS-CG for noninvasive ratiometric fluorescence imaging of endogenous H_2_S generation in the HCT116 tumors of mice. When the HS-CG was i.v. injected into mice, the fluorescence in the IR-HS channel was significantly enhanced, and bright fluorescence was displayed in the ICG channel (Figure 5d,e); the fluorescence ratio F_IR-HS_/F_ICG_ was determined to be ~0.52 ± 0.06 (Figure 5f). The fluorescence of the tumor IR-HS was remarkably increased (∼1.9-fold) with intratumoral (i.t.) injection of _L_-Cys, while the fluorescence of ICG showed no obvious change. The F_IR-HS_/F_ICG_ of the tumors increased ~1.9-fold after 12 h, which was obviously reduced when the CSE activity was inhibited by PAG and AOAA (Figure 5f). These results demonstrate that HS-CG could reliably be used in the ratiometric imaging of tumors with endogenous H_2_S levels in living mice.

## 3. Materials and Methods

### 3.1. Materials and Instrumentation

All chemicals were acquired from Sigma-Aldrich (Shanghai, China) unless otherwise stated. DSPE-PEG2000 was purchased from Avanti (Alabaster, AL, USA). A 3-(4,5-dimethylthiazol-2-yl)-2,5-diphenyltetrazoliumbromide (MTT) Kit was bought from KeyGen Biotech. Co., Ltd., (Nanjing, China). DPPC was bought from Sigma-Aldrich. ICG was acquired from MedChemExpress (Shanghai, China). IR-HS was synthesized according to our previously reported synthesis method [38].

Absorption and fluorescence spectra were performed on an Ocean Optics UV–Visible spectrometer (Ocean Optics, Dunedin, FL, USA) and HORIBA Jobin Yvon Fluoromax-4 fluorescence spectrometer (HORIBA Jobin Yvon, Paris, France), respectively. Fluorescence images were acquired on an Olympus IX73 fluorescent inverted microscope (Olympus LS, Tokyo, Japan). Dynamic light scattering (DLS) was conducted on a 90 Plus/BI-MAS equipment (Brookhaven, New York, NY, USA). Transmission electron microscopy (TEM) analysis was performed on a JEM-1011 transmission electron microscope (JEOL, Ltd., Tokyo, Japan). The MTT analysis was conducted on a microplate reader (Tecan, Grödig, Austria). The fluorescence images in mice were collected using an IVIS Lumina XR III system (PerkinElmer, Waltham, MA, USA).

### 3.2. Preparation of Liposome Nanoprobe HS-CG

The liposome nanoprobe HS-CG was prepared through a thin-film hydration method. Briefly, IR-HS (0.027 mg), ICG (0.035 mg), DPPC (4 mg), cholesterol (2 mg), and DSPE-PEG2000-OMe (4 mg) were dissolved in a mixture of CH_3_CH_2_OH (1 mL) and DMSO (5 μL) to form a homogeneous solution. Then, we removed the organic solvent to form a thin layer by rotary evaporator. Subsequently, we added D.I. water (9 mL) into a thin layer under sonication. The remaining aqueous solution was further sonicated for 5 min and then passed through 0.22 μm filters three times. Finally, the heterogeneous liposome nanoprobe HS-CG was formed. The HS-CG was washed three times under ultrafiltration. The final solution was kept in the dark at 4 °C.

### 3.3. Investigation of the Sensitivity and Selectivity of HS-CG toward H_2_S

To evaluate the sensitivity of HS-CG toward H_2_S in solution, HS-CG (5/5 μM IR-HS/ICG) was incubated with different concentrations of NaHS (0, 1, 2, 5, 10, 15, 20, 50, 80, 100, 150, 200, 300, 400, 500 μM) in PBS buffer (pH 7.4) at 37 °C for 6 min. The fluorescence spectra of ICG within HS-CG were collected using the HORIBA Jobin Yvon Fluoromax-4 fluorometer (λ_ex_ = 780 nm, λ_em_ = 800–950 nm). The fluorescence spectra of IR-HS within HS-CG were collected using the HORIBA Jobin Yvon Fluoromax-4 fluorometer (λ_ex_ = 680 nm, λ_em_ = 700–800 nm). The fluorescence intensities at 728 and 816 nm were obtained from the fluorescence spectra. The fluorescence intensity ratio (F_728_/F_816_) was applied to calculate the detection limit according to the formula (3δ/*k*), where δ is the standard deviation of F_728_/F_816_ of the blank signal; and *k* is the slope between the F_728_/F_816_ versus H_2_S concentration.

To investigate the selectivity of HS-CG toward H_2_S, HS-CG (5/5 μM IR-HS/ICG) was incubated with reducing agents (e.g., 500 μM NaHS, 5 mM glutathione (GSH), 1 mM ascorbic acid (VC), 1 mM _L_-cysteine (_L_-Cys), 1 mM homocysteine (Hcy)) at 37 °C for 6 min. The fluorescence spectra of ICG within HS-CG were collected using the HORIBA Jobin Yvon Fluoromax-4 fluorometer (λ_ex_ = 780 nm, λ_em_ = 800–950 nm). The fluorescence spectra of IR-HS within HS-CG were collected using a HORIBA Jobin Yvon Fluoromax-4 fluorometer (λ_ex_ = 680 nm, λ_em_ = 700–800 nm).

### 3.4. Cell Culture

Colon cancer cells (HCT116 cells) were grown in McCoy′s 5A medium. The medium was supplemented with 10% (*v*/*v*) fetal bovine serum (FBS) and 100 units/mL penicillin plus 100 μg/mL streptomycin. HCT116 cells were kept in a humidified incubator containing 5% CO_2_ at 37 °C.

### 3.5. Ratiometric Imaging of H_2_S in Living Tumor Cells

HCT116 cells were inoculated in a glass-bottom plate. After 12 h, the cells were incubated with HS-CG (4/4 μM IR-HS/ICG) for 3 h and washed with PBS buffer (1×), and then incubated with or without NaHS (1 mM) for 1 h. To eliminate the endogenous H_2_S, cells were pretreated with ZnCl_2_ (300 μM) for 10 min. To eliminate the exogenous H_2_S, cells were pretreated with ZnCl_2_ (300 μM) for 10 min, and cells were treated with NaHS (1 mM) for 1 h. To increase the endogenous H_2_S level, HCT116 cells were treated with _L_-Cys (200 μM) for 1 h. To inhibit the cystathionine γ-lyase (CSE) and cystathionine b-synthase (CBS) activity, cells were pretreated with DL-propargylglycine (PAG, 50 mg/L) plus aminooxyacetic acid (AOAA, 20 μΜ) for 0.5 h, and then treated with _L_-Cys (200 μM) for 1 h. After each treatment, cells were incubated with HS-CG (4/4 μM IR-HS/ICG) for 3 h. The fluorescence images were acquired on an Olympus IX73 fluorescent inverted microscope.

### 3.6. Animals and Tumor Models

Female BALB/c nude mice (body weight: ~20 g) were purchased from Gempharmatech Co. Ltd. (Nanjing, China). All animal experiments complied with the Institutional Animal Care and Use Committee (IACUC). To establish the tumor model, HCT116 cells (1.0 × 10^6^ cells) suspended in 25 μL McCoy′s 5A and 25 μL Matrigel were inoculated in the right hind-leg region of the mice. The tumors were grown to 100–200 mm^3^ before further application.

### 3.7. Ratiometric Fluorescence Imaging of Exogeneous H_2_S in Mice

To image the exogeneous H_2_S level in mice, saline (25 μL), NaHS (1 mM, 25 μL), or NaHS (1 mM, 25 μL) containing ZnCl_2_ (1 mM) was subcutaneously injected into the right hind-leg region of the mice. Subsequently, HS-CG (20/20 μM IR-HS/ICG, 25 μL) in saline was subcutaneously injected into the mice. After 15 min, the fluorescence images of the mice were acquired on an IVIS Lumina XR III imaging system using a 660 nm excitation filter with a 730–790 nm emission filter for imaging IR-HS within HS-CG and a 780 nm excitation filter with n 825–865 nm emission filter for imaging ICG within HS-CG.

### 3.8. Ratiometric Imaging of Endogenous H_2_S Tumor of Mice

To image H_2_S in the HCT116 tumors, mice bearing subcutaneous HCT116 tumors were intravenously (i.v.) injected with HS-CG (40/40 μM IR-HS/ICG, 200 μL). After injection, the mice were i.t. injected with saline (25 μL) or _L_-Cys (1 mM, 25 μL) or _L_-Cys (1 mM, 25 μL) containing PAG (5 mg/kg) +AOAA (5 mg/kg). The fluorescence images were acquired at 12 h on an IVIS Lumina XR III imaging system using a 660 nm excitation filter with a 730–790 nm emission filter for imaging IR-HS within HS-CG and a 780 nm excitation filter with an 825–865 nm emission filter for imaging ICG within HS-CG.

## 4. Conclusions

In conclusion, we developed a ratiometric NIR fluorescence liposome nanoprobe (HS-CG) for H_2_S detection in vivo. We showed that HS-CG displayed fast kinetics toward H_2_S because DPPC that was positively charged could capture the anion HS^−^ to increase the local concentration of HS^−^ around HS-CG and improve the reaction rate between HS-CG and H_2_S. Doping positively charged DPPC into the nanoparticles could provide an effective strategy to improve the response toward H_2_S and enhance the detection sensitivity of H_2_S. HS-CG offers a turn-on NIR fluorescence ratio F_728_/F_816_ that could be successfully applied to monitor H_2_S levels in living HCT116 cells and tumor H_2_S levels in mice, exhibiting a highly fast response that is specific toward H_2_S. Thus, the H_2_S-activatable ratiometric NIR fluorescence liposome nanoprobe has the potential to serve as an efficient imaging nanoprobe for the detection of H_2_S-related physiological and pathological processes in vivo.

## Figures and Tables

**Figure 1 molecules-28-01898-f001:**
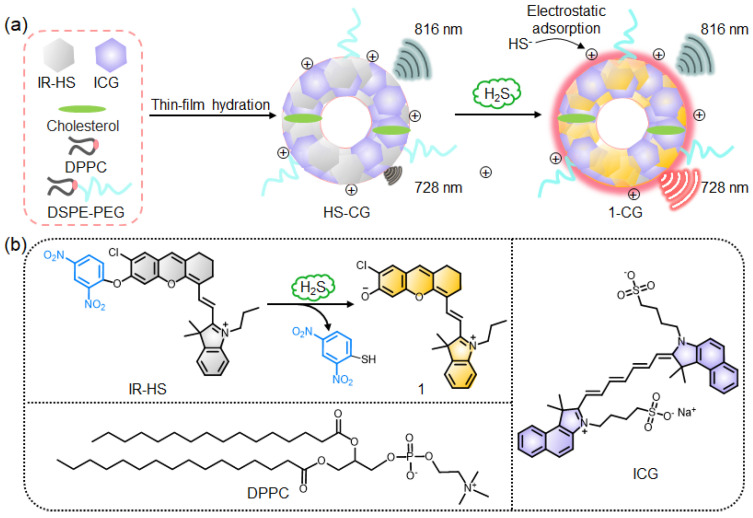
General design of the H_2_S-activatable NIR ratiometric fluorescence probe HS-CG. (**a**) Cartoon illustrates the preparation of HS-CG and the mechanism of HS-CG for the detection of H_2_S. (**b**) Proposed chemical conversion of IR-HS into 1 upon reduction by H_2_S, and the chemical structures of DPPC and ICG. HS-CG was prepared as liposome nanoparticles via DSPE-PEG2000- and DPPC-assisted encapsulation of IR-HS and ICG.

**Figure 2 molecules-28-01898-f002:**
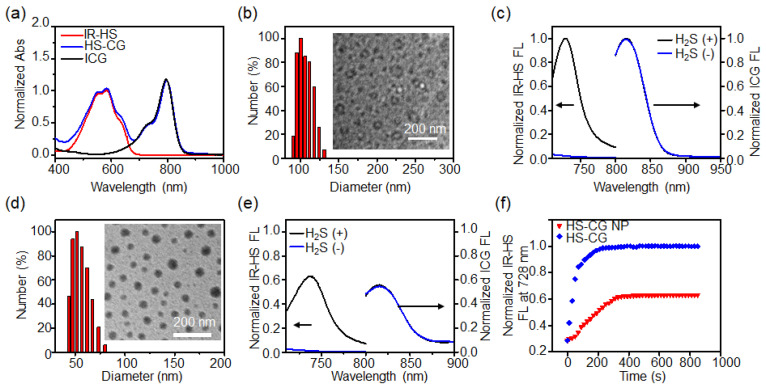
Characterization of HS-CG in vitro. (**a**) UV–Vis–NIR absorption spectra of HS-CG, IR-HS, and ICG NPs in PBS buffer. (**b**) Dynamic light scattering (DLS) analysis and transmission electron microscopy (TEM) image (inset) of HS-CG. (**c**) Fluorescence spectra of HS-CG before (−) and after (+) reaction with NaHS (500 μM) for 30 min. (**d**) DLS analysis and TEM image (inset) of HS-CG NP, and (**e**) fluorescence spectra of HS-CG NP before (−) and after (+) reaction with NaHS (500 μM) for 30 min. For a comparison with the reaction rate and fluorescence intensity of the prepared liposome nanoparticles HS-CG, we prepared nanoparticles HS-CG NP via the DSPE-PEG2000-assisted encapsulation of IR-HS and ICG. (**f**) Time-dependent fluorescence intensity change at 728 nm of HS-CG and HS-CG NP before (−) and after (+) reaction with NaHS (1 mM).

**Figure 3 molecules-28-01898-f003:**
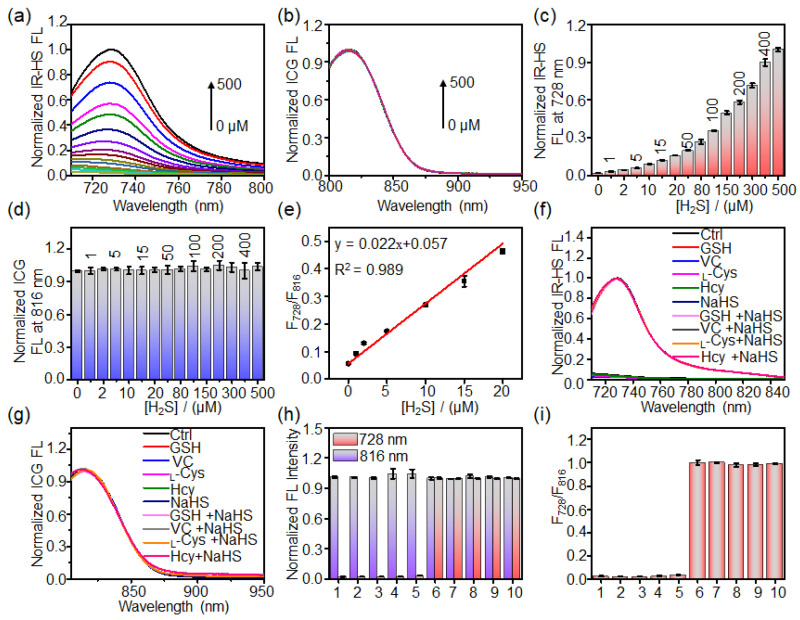
Study on the reactivity of HS-CG with H_2_S. (**a**) Fluorescence spectra of IR-HS and (**c**) change in the fluorescence intensity at 728 nm of IR-HS within HS-CG (5/5 μM IR-HS/ICG) upon incubation with NaHS (0, 1, 2, 5, 10, 15, 20, 50, 80, 100, 150, 200, 300, 400, 500 μM) in PBS buffer (pH 7.4) at 37 °C for 6 min. (**b**) Fluorescence spectra of ICG and (**d**) change in the fluorescence intensity at 816 nm of ICG within HS-CG (5/5 μM IR-HS/ICG) upon incubation with NaHS (0, 1, 2, 5, 10, 15, 20, 50, 80, 100, 150, 200, 300, 400, 500 μM) in PBS buffer (pH 7.4) at 37 °C for 6 min. (**e**) The linear relationship between the F_728_/F_816_ and the concentration of NaHS from 0 to 20 μM. Fluorescence spectra of (**f**) IR-HS and (**g**) ICG within HS-CG (5/5 μM IR-HS/ICG), (**h**) fluorescence intensities of HS-CG at 728 and 816 nm, and (**i**) F_728_/F_816_ of HS-CG upon incubation with different reagents (1—Ctrl, 2—5 mM glutathione (GSH), 3—1 mM ascorbic acid (VC), 4—1 mM L-cysteine (_L_-Cys), 5—1 mM homocysteine (Hcy), 6—500 μM NaHS, 7—5 mM GSH plus 500 μM NaHS, 8—1 mM VC plus 500 μM NaHS, 9—1 mM _L_-Cys plus 500 μM NaHS, 10—1 mM Hcy plus 500 μM NaHS) at 37 °C for 6 min. Values are mean ± standard deviation (s.d.) (*n* = 3).

**Figure 4 molecules-28-01898-f004:**
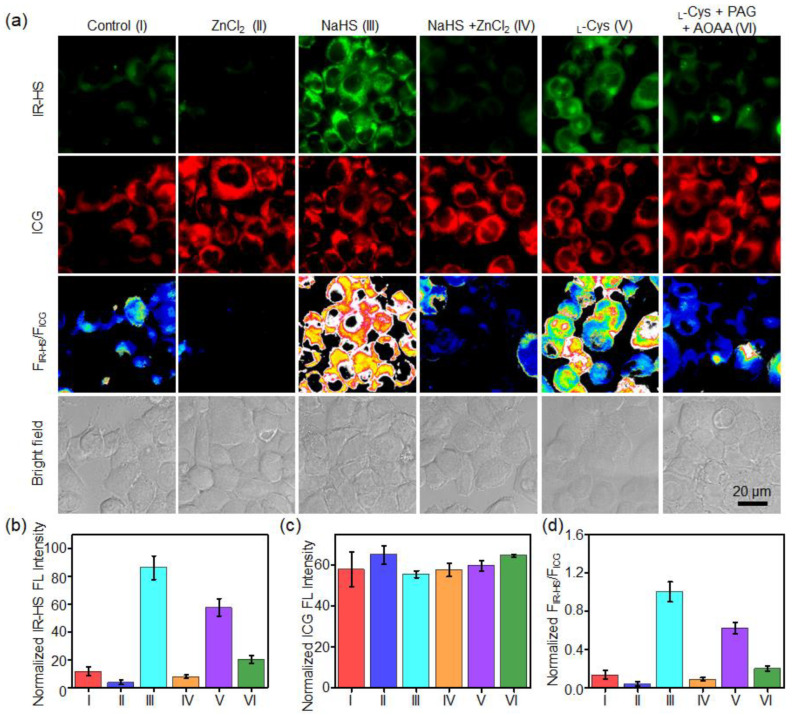
Ratiometric imaging of H_2_S in living HCT116 cells. (**a**) Fluorescence and ratiometric images, intensities of IR-HS (**b**) and ICG (**c**) of HCT116 cells incubated with HS-CG and the indicated reagents. Cells were untreated (Control) or pretreated with ZnCl_2_ (300 μM, 10 min), NaHS (1 mM, 1 h), ZnCl_2_ (300 μM, 10 min), plus NaHS (1 mM, 1 h), _L_-Cys (200 μM, 1 h), or PAG (50 mg/L, 0.5 h) + AOAA (20 μΜ, 0.5 h) + _L_-Cys (200 μM, 1 h), and incubated with HS-CG (4/4 μM IR-HS/ICG) for 3 h. (**d**) The fluorescence ratio (F_IR-HS_/F_ICG_) of the HCT116 cells after treatment with the indicated conditions in (**a**). Values are the mean ± s.d. (*n* = 3).

**Figure 5 molecules-28-01898-f005:**
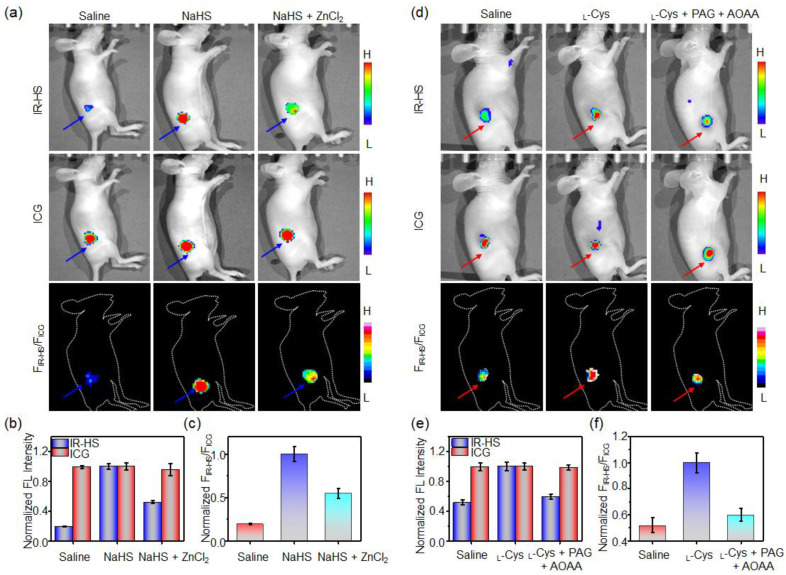
Non-invasive imaging of H_2_S in vivo. (**a**) Fluorescence and ratiometric images of mice following the subcutaneous injection of saline (25 μL) plus HS-CG (20/20 μM IR-HS/ICG, 25 μL), HS-CG plus NaHS (1 mM, 25 μL), and HS-CG plus ZnCl_2_ (1 mM) + NaHS (1 mM, 25 μL). (**b**) Fluorescence intensities of HS-CG and (**c**) the fluorescence intensity ratio (F_IR-HS_/F_ICG_) following treatment with HS-CG, HS-CG plus NaHS, and HS-CG plus NaHS plus ZnCl_2_. (**d**) Fluorescence and ratiometric images of HCT116 tumors in mice at 12 h following intravenous (i.v.) injection of HS-CG (40/40 μM IR-HS/ICG, 200 μL), HS-CG plus _L_-Cys (1 mM, 25 μL), and HS-CG plus _L_-Cys plus PAG (5 mg/kg) +AOAA (5 mg/kg). (**e**) Fluorescence intensities of HS-CG and (**f**) the fluorescence intensity ratio (F_IR-HS_/F_ICG_) following treatment with the indicated treatment. Values are mean ± s.d. (*n* = 3).

## Data Availability

The data that support the findings of this study are available from the corresponding author upon reasonable request.

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
