# Peer review of "Ratiometric Near-Infrared Fluorescence Liposome Nanoprobe for H2S Detection In Vivo"

_molecules, 2023, doi:10.3390/molecules28041898_

Round 1
Reviewer 1 Report
The manuscript titled “Ratiometric near-infrared fluorescence liposome nanoprobe for H2S detection in vivo” by Wu et al. presented a H2S-activatable near-infrared ratiometric fluorescence liposome nanoprobe HS-CG for accurate imaging in vivo. This work provided a new strategy of electrostatic adsorption to enhance the reaction rate of the nanoprobe and H2S, which could serve as a platform to improve the sensitivity of H2S imaging. This article is suitable for publication in Molecules after addressing some minor issues.
1. The “DSPE” was not defined throughout the manuscript.
2. The captions in Figure 2a and b did not match the contents.
3. Nanoprobe stability plays a key role in fluorescence imaging in vivo. Thus, stability study on HS-CG liposome should be performed.
4. For selectivity of liposome nanoprobe HS-CG, mixture sample containing NaHS should also be used for detection.
5. The manuscript had some language and format issues, authors should check it carefully. For example:
(1) Figure 3d, the thickness of the error bars was inconsistent;
(2) Format issue of References 10 and 17.
Author Response
We appreciate the reviewer’ comments and suggestions. In the revised manuscript, all these comments have been addressed with point-by-point response as shown in the attachment. Please see the attachment.

Reviewer 2 Report
In this manuscript, An and co-workers are reporting in interesting ratiometric fluorescence probe for in vivo detection of H2S in living systems. The overall layout of this manuscript seems to be reasonable. Therefore, I suggest considering this manuscript after following revisions/suggestions.
(1). The ratiometric fluorescence sensors which especially Turn ON fluorescence in the NIR range are highly useful in biological applications. Therefore, I would suggest authors to extend their introduction by briefly discussing the advantages of such probes while citing some of the recent references. For example: Sensors and Actuators B: Chemical, 343, 130063, New Journal of Chemistry, 45(20), 9102-9108, Sensors and Actuators B: Chemical, 371, 132512, Analytical chemistry, 92(8), 6111-6120.
(2). Authors should provide LOD calculations in the supporting information. In addition, this LOD value must be compared with previously reported probes.
(3). Authors must comment of the approximate response time as well as stability of the fluorescent probe.
(4). Probe must be studied under different pH conditions to analyze the behavior.
(5). The fluorescence quantum yield, Stokes shift, absorbance, emission and molar absorptivity should be calculated for the probe in different solvent s and must be provided in a table form.
(6). Cell viability results must be provided for the probe.
Author Response

(The authors gave the same response as above.)

Round 2
Reviewer 2 Report
Accept in the current form.